# Epidemic modelling suggests that in specific circumstances masks may become more effective when fewer contacts wear them
Peter Klimek [1,2,3,4] ✉, Katharina Ledebur[1,2] & Stefan Thurner [1,2,5]

## Abstract

**Background** The effectiveness of non-pharmaceutical interventions to control the spread of SARS-CoV-2 depends on many contextual factors, including adherence. Conventional wisdom holds that the effectiveness of protective behaviours, such as wearing masks, increases with the number of people who adopt them. Here we show in a simulation study that this is not always true.

**Methods** We use a parsimonious network model based on the well-established empirical facts that adherence to such interventions wanes over time and that individuals tend to align their adoption strategies with their close social ties (homophily).

**Results** When these assumptions are combined, a broad dynamic regime emerges in which the individual-level reduction in infection risk for those adopting protective behaviour increases as adherence to protective behaviour decreases. For instance, at 10 % coverage, we find that adopters face nearly a 30 % lower infection risk than at 60 % coverage. Based on surgical mask effectiveness estimates, the relative risk reduction for masked individuals ranges from 5 % to 15 %, or a factor of three. This small coverage effect occurs when the outbreak is over before the pathogen is able to invade small but closely knit groups of individuals who protect themselves.

**Conclusions** Our results confirm that lower coverage reduces protection at the population level while contradicting the common belief that masking becomes ineffective at the individual level as more people drop their masks.

## Plain Language Summary

Face masks have been used as one tool to protect people against COVID-19 infection. Here, we perform mathematical simulations to investigate how well mask-wearing works in different scenarios. Counterintuitively, our simulations showed that as fewer people follow these measures, the risk of infection decreases for those who still do. For instance, when only 10% of people follow them, their infection risk gets reduced by almost 30% compared to situations where 60% follow. Our findings challenge the idea that masks become ineffective when fewer people wear them. The overall public health benefit still increases as more people wear masks, but their protective effect at the individual level can still be substantial if only a few people wear them.

The SARS-CoV-2 pandemic led to the widespread and sustained implementation of a number of non-pharmaceutical interventions (NPIs) such as social distancing, wearing face masks, or test-trace-isolate strategies[1–3]. From a scientific viewpoint, this provided a unique opportunity to accumulate real-world evidence on the effectiveness of such NPIs[4–7]. Meanwhile, there is overwhelming empirical evidence that many of these interventions do indeed curb the virus spread[8,9]. The literature is less unanimous when it comes to precise quantitative estimates of the effectiveness of NPIs[5,7,8,10]. The heterogeneity in these estimates can be explained by a plethora of contextual factors impacting their adoption,

ranging from geographical, cultural, socio-economic and health care and behaviour factors[11–14].

The extent to which individuals adhere to an NPI has been shown to change significantly over time, even when there has been no formal change in the tightening or loosening of NPIs[15]. Adherence to physical distancing measures has been observed to wane over time[15–18]. Adherence to low-cost habituation measures, such as wearing masks, increased during the early phases of the pandemic, likely linked to extensive public health messaging[16,19,20]. However, even with these low-cost measures adherence plateaued and started to decline after vaccines were rolled out, the Omicron

[1]Section for Science of Complex Systems, Medical University of Vienna, Vienna, Austria. [2]Complexity Science Hub Vienna, Vienna, Austria. [3]Supply Chain Intelligence Institute Austria, Vienna, Austria. [4]Division of Insurance Medicine, Karolinska Institutet, Stockholm, Sweden. [5]Santa Fe Institute, Santa Fe, NM, USA. ✉e-mail: peter.klimek@meduniwien.ac.at

variant became dominant and an increasing number of countries dropped mask mandates[21–24].

Further, adherence to NPIs is strongly related to homophily[25]. It is a well-established fact that individual behaviour and attitudes cluster in social networks, meaning that attitudes of a person often conform with the attitudes of its close social contacts[26,27]. Homophily is generally observed in health behaviour[28,29] and was recently confirmed in the context of the COVID-19 pandemic[30,31]. For instance, the likelihood of an individual wearing a mask is closely related to the frequency of actual mask use, both in the general population, and in close social contacts[32]. A study evaluating mask-wearing status of pairs of people using public livestream footage from webcams across seven US cities in September 2020 found that 83% of the participants had the same status than the person they had contact with[33]. In 41% of cases, both people were wearing masks, and in 42% of cases, both were not, confirming strong assortative mixing patterns in social contexts. Observational studies in the Czech Republic confirmed these observations by showing that mask-wearing status of other customers in a shop was one of the strongest predictors for mask-wearing behaviour of new customers[34].

For behavioural interventions with imperfect (incomplete) adherence, two types of effectiveness of NPIs can be distinguished. First, by population-level effectiveness we refer to the overall reduction of infections when an NPI is adopted by a certain fraction of the population compared to a situation in which the NPI is adopted by fewer people (e.g., zero) but everything else stays the same. Second, for the individual-level effectiveness we compare the infection risk for an individual that adopts an NPI to the risk of an individual not adopting the NPI within the same population. Studies that quantify effectiveness by comparing the number of infections in regions with and without an NPI seek to estimate population-level effectiveness, as it is often the case in ecological study designs or studies relying on natural experiments[4–7]. Case-control studies[35,36] or randomised controlled trials[37] often measure individual-level effectiveness.

Epidemiological models make it clear that individual-level and population-level effectiveness are very different concepts. It is well-established that if the rate of adoption (coverage rate) and the effectiveness of NPIs are high enough, epidemic outbreaks can be successfully suppressed, meaning that a primary infection will lead to (on average) fewer than one secondary cases[38–40]. In this case of suppression an NPI can have a high population-level effectiveness combined with low individual-level effectiveness, as non-adopters are indirectly protected through the overall low infection risk. With lower coverage, where outbreaks are not suppressed, the population-level effectiveness may be low, but the individual-level effectiveness may be much higher than in situations where outbreaks are suppressed. In other words, the effectiveness of an NPI at the individual level (i.e. how much wearing a mask reduces one's personal risk of infection) depends on population-level characteristics such as the proportion of the total population that adopts the NPI.

In this paper, we use a simulation study to show that an additional factor can modulate the effectiveness of a wide range of NPIs, such as the wearing of face masks. We consider the broad class of NPIs that may or may not be adopted by individuals and that reduce the likelihood of transmission when adopted by either the transmitting or the receiving individual. Obviously, face masks are of this type but also other habituation measures such as keeping distance can reduce the chance of transmission, independent from whether the adopting individual is the transmitter or receiver.

Here we aim to understand how the individual- and population-level effectiveness of mask-wearing depends on the coverage rate. To put it more colloquially, if my own benefit from wearing a mask increases with the number of others wearing a mask, should I wear one, if no-one else is wearing one? We study this question in a simple epidemiological model that acknowledges two repeatedly confirmed properties regarding the adoption of NPIs that were observed throughout the pandemic. First, we assume that NPI adherence wanes over time; someone who initially wore a mask during almost all non-household contacts will, over time, wear the mask at fewer and fewer occasions. Second, we assume a certain degree of behavioural homophily, in the sense that the adoption behaviour of NPIs clusters in

social networks: individuals with close social ties tend to synchronise their individual decisions on protection measures. In all other aspects, we opt for a parsimonious modelling approach so that we can clearly isolate the effects originating from these assumptions and how they play out on the relation between individual- and system-level effectiveness of NPIs.

Under these assumptions, we observe a counterintuitive dynamic regime, where the infection risk for individuals adopting protective behaviour decreases as adherence to these measures wanes. This phenomenon arises due to the, to the best of our knowledge, previously unknown, small coverage effect, where the simulated outbreak concludes before transmission occurs among individuals who adopt protective behaviour.

## Methods
### Homophily
In the model, all individuals have a static state that remains fixed throughout the simulations, indicating whether or not they are willing to adopt a protective behaviour. Over time, however, the probability by which they will actually adopt this behaviour decreases over time to implement waning of NPI adherence. Each individual is either susceptible, infected or recovered[41]. To include homophily, epidemic spreading takes place on a social contact network that shows assortative mixing with respect to the adoption of protective behaviour. Assortative mixing is implemented with a small-world network model, see Fig. 1[42]. In this network, $N$ nodes are positioned on a ring and are linked to their $k$ nearest neighbours, resulting in a strongly clustered network, i.e., two neighbours of a given node are also likely to be connected. With a certain probability, $\epsilon$, links are randomly rewired. The parameter $\epsilon$ reflects a random linking between (family, work, and leisure) groups within the society. $\epsilon = 0$ means a regular structure where everyone is linked to exactly $k$ neighbours; $\epsilon = 1$ leads to a random network in which each pair of nodes is connected with the same probability. Finally, to introduce homophily in the adoption of protective behaviour, we first group those individuals (nodes) with the same behaviour on the ring and then randomly swap the behaviour state between any two nodes with probability $1 - \eta$, see Fig. 1. $\eta$ acts as a control parameter for the degree of homophily; large values represent a behaviourally clustered society, small values lead to a non-homophilic situation.

### Transmission reduction
In the model, the paradigmatic SIR-type dynamics unfold on networks of this type. Initially, all nodes are in the susceptible state except for a small fraction of nodes in the infected state. At each timestep a susceptible node can be infected by an infected neighbouring node with probability, $\alpha$, if both nodes don't adopt protective behaviour. Adoption of protective behaviour reduces the probability of a successful transmission by a time-dependent factor, $q_r(t)$, for the receiving individual and by $q_s(t)$ for the transmitter (source). We assume that $q_r(t)$ and $q_s(t)$ decrease at every timestep by a constant percentage to reflect waning of NPI adherence. Nodes remain infected for $1/\gamma$ timesteps, after which they recover and remain in this state for the rest of the simulation.

We measure how likely adopting and non-adopting individuals get infected during an outbreak and demonstrate that individual- and population-level infection risk depend on the coverage rate in a non-trivial way.

### Network
We consider a small world network of the Watts-Strogatz type with $N$ nodes positioned on a ring where they are connected to their $k$ nearest neighbours. With probability, $\epsilon$, a link is chosen, disconnected from one of its randomly chosen end nodes and rewired to any randomly chosen other node of the network (self-links are excluded). The resulting network can be described by an $N \times N$ adjacency matrix, $A$, with entries, $A_{ij} = 1$, if nodes $i$ and $j$ are connected (neighbouring) and $A_{ij} = 0$, otherwise. Every node $i$ has a static state, $M_i$, signalling whether it adopts protective behaviour ($M_i = 1$) or not ($M_i = 0$) at the start of the simulation. The coverage rate, $m$, represents the probability that a node adopts protective behaviour, $\sum_i M_i = mN$. The

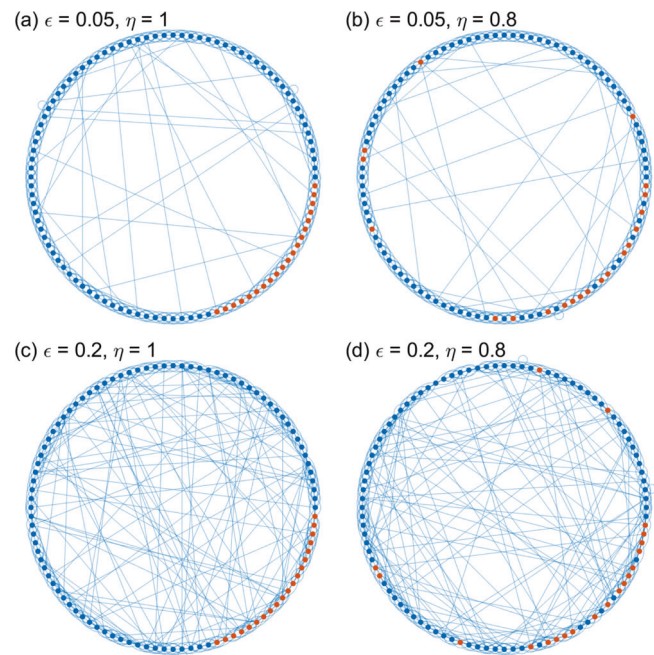

(a) $\epsilon = 0.05$, $\eta = 1$ (b) $\epsilon = 0.05$, $\eta = 0.8$

(c) $\epsilon = 0.2$, $\eta = 1$ (d) $\epsilon = 0.2$, $\eta = 0.8$

**Fig. 1 | Illustration of a small-world network with assortative adoption of protective behaviour (homophily).** We show $N = 100$ individuals (nodes) out of which 20 adopt (red) and 80 do not adopt the behaviour (blue). Nodes are positioned on a ring and linked to their $k = 6$ nearest neighbours; individuals with the same behaviour cluster in the network. To reflect imperfect assortative mixing, with some probability ($\epsilon$) a link is chosen and rewired toward a randomly chosen different node and with another probability $(1-\eta)$ the behaviour state (colour) is swapped between two randomly chosen nodes. The resulting networks are illustrated for ($\epsilon,\eta$) being (**a**) (0.05, 1), (**b**) (0.05, 0.8), (**c**) (0.2, 1) and (**d**) (0.2, 0.8).

homophily parameter, $\eta$, is the fraction of nodes that do assortatively mix with each other. That is, a number of $\eta m N$ nodes $i$, with $M_i = 1$, are grouped next to each other on the network, whereas a fraction of $(1 - \eta)m N$ of nodes with $M_i = 1$ is randomly distributed across the ring, see Fig. 1.

## Simulation

Each node, $i$, is endowed with a dynamic epidemic state variable, $X_i(t)$, which takes values in $X_i(t) \in \{S, I, R\}$ corresponding to the states in the SIR model: susceptible, infected, or recovered. At every timestep, a susceptible individual $i$ can be infected with probability $\alpha$ by any of its neighbouring infected individuals $j$ if neither of the two adopts protective behaviour ($M_i = M_j = 0$). If $M_i = 1$, the infection risk decreases by a factor of $1 - q_r(t)$, if $M_j = 1$, the risk decreases by $1 - q_s(t)$. Adherence to these behaviours decreases exponentially with a rate $0 < q \le 1$, such that $q_r(t) = q_r(0)q^t$ and $q_s(t) = q_s(0)q^t$, respectively. Infected individuals recover with probability $\gamma$.

At every timestep, $t$, the model loops over all nodes $i = 1, \ldots, N$ and performs a state update according to the following protocol (parallel update).

- If $X_i(t) = S$, identify the set of all close infectious contacts of $i$, $Nb(i)$, as $Nb(i) = \{j | A_{ij} = 1 \wedge X_j(t) = I\}$. For each node $j \in Nb(i)$, set $X_i(t+1) = I$ with probability, $\alpha \cdot (1 - q_r(t))^{M_i} \cdot (1 - q_s(t))^{M_j}$.
- If $X_i(t) = I$, set $X_i(t+1) = R$ with probability $\gamma$.
- Proceed with the next node. Once all nodes have been updated at time $t$, proceed to the next timestep, $t + 1$.

We denote the number of new cases by $C(t, M = 1)$ and $C(t, M = 0)$, for those that do and do not adopt protective behaviour, respectively.

## Observables

We measure the risk of becoming infected during the outbreak on an individual and on a population scale. While the individual infection risk

gives the probability that an individual will become infected as a function of whether or not they adopt protective behaviour, the population-level infection risk measures the total number of infections in the whole population. The individual infection risks in the two groups are

$$IIR(M = 1) = \frac{\sum_t C(t, M = 1)}{mN} \tag{1}$$

and

$$IIR(M = 0) = \frac{\sum_t C(t, M = 0)}{(1 - m)N}, \tag{2}$$

respectively; the population infection risk is given by

$$PIR = \frac{\sum_{t,M} C(t, M)}{N}. \tag{3}$$

## Parameters

Unless stated otherwise, the following parameter settings were used. Simulations were performed for $N = 10^5$ nodes and averaged over 20 independent iterations. The parameters were chosen such that $k = 20$, $\alpha = 0.02$, $\gamma = 0.1$, $q = 0.99$, $q_r(0) = q_s(0) = 1$, $\eta = 1$, and $\epsilon = 0.1$. For the initial condition we randomly choose ten nodes and set their $X_i(t = 0) = I$; for all the others we set $X_i(t = 0) = S$. The model halts once the outbreak has ended, i.e., $X_i(t) \ne I$ for all $i$.

## Reporting summary

Further information on research design is available in the Nature Portfolio Reporting Summary linked to this article.

## Results

Example runs for different coverage are shown in Fig. 2a–c. We show the averages and the 68% confidence intervals (CI) of the number of new cases as a function of time over multiple iterations for increasing coverage, $m$, i.e., increasing adoption of protective behaviour. For individuals $i$ that do not adopt protective behaviour ($M_i = 0$), the height of the epidemic peaks decreases with increasing coverage as the source-controlling effects indirectly protect them. The duration of the outbreak increases however, leading to a situation where the total infection risk for non-adopting individuals during the outbreak changes little for sufficiently low levels of $m$, see the corresponding plateau for low $m$ in Fig. 2d.

Intriguingly, for adopting individuals the individual infection risk, *IIR,* may exhibit a pronounced maximum at an intermediate coverage rate. Simulation runs for different values of coverage ($m$) show comparable heights of the epidemic peak for adopting individuals; see Fig. 2a–c. However, the duration of this peak increases substantially with increasing $m$. As a result, there is a regime where the individual infection risk for adopting individuals decreases with decreasing coverage, as shown in Fig. 2d. For the settings used, 61% of the adopting individuals get infected during the outbreak if they make up only 10% of the population, but 78% get infected if they account for 60% of the total population. We refer to this phenomenon as the small coverage effect.

On a population level, this small coverage effect reveals itself in the form of a plateau in the population infection risk, *PIR,* at intermediate coverage, see Fig. 2e. On this plateau the *PIR*-increasing small coverage effect approximately cancels with the increase in the number of adopting individuals with higher *IIR.* For reference, Fig. 2e also shows results for the *PIR* without waning in which the monotonous decrease in *PIR* under increasing coverage is recovered.

Different mechanisms are at work that reduce the individual infection risk for adopting individuals when coverage becomes very high or low, respectively. Starting from a coverage of 80% or more, simulation runs start to occur in which the epidemic dies out before adherence has waned

**Fig. 2 | Demonstration of the small coverage effect of protective behaviour.** The number of new cases over time for those adopting protective behaviour ($C(t, M = 1)$, red) and those who do not ($C(t, M = 0)$, blue) are shown for coverage from (**a**) $m = 0.1$ over (**b**) $m = 0.3$ to (**c**) $m = 0.5$; shaded areas denote the 68% CI. Insets illustrate the networks for these settings, respectively. Results for the (**d**) individual infection risk, $IIR(M = 0, 1)$, as a function of coverage $m$ are shown as solid lines and compared to a simulation for which mask-wearing adherence does not wane (dotted lines, using $q = 1$ and $q_r(0) = q_s(0) = 0.4$); error bars denote a standard deviation. If adherence wanes, the infection risk increases with $m$ for $m < 0.6$; for higher values of $m$ the distribution of outbreak sizes becomes bimodal (marker size is proportional to the fraction of simulation runs in each mode). The infection risk is always a monotonously decreasing function of $m$ if adherence does not wane. On the (**e**) population level we observe a plateau with similar risks, $PIR$, at intermediate coverage next to the bifurcation for high coverage.

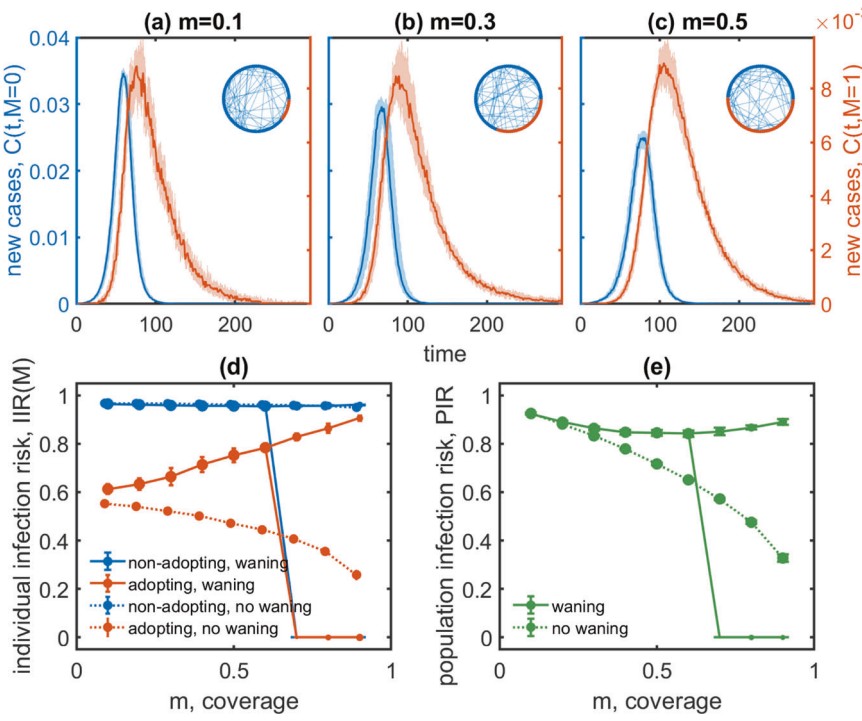

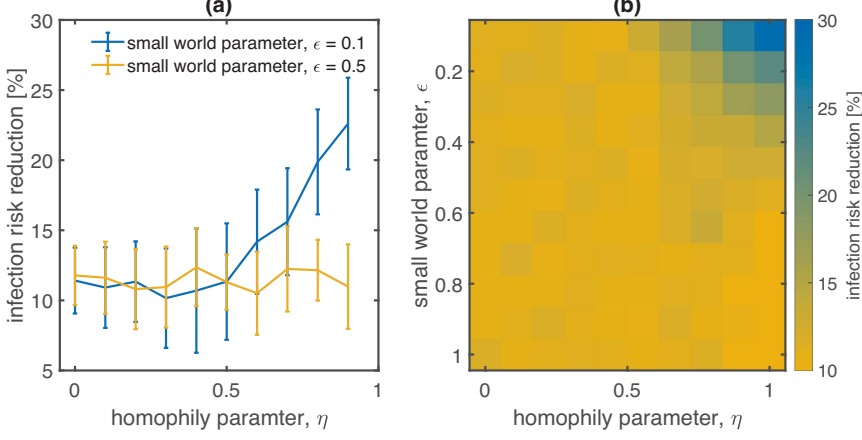

**Fig. 3 | Homophily amplifies the small coverage effect.** We show the reduction of the individual infection risk [%] due to the small coverage effect from a coverage of of 60% to 10%. We find (**a**) an infection risk reduction of about 20–25% for small world parameters representing strong homophily (large $\eta$). The reduction decreases with decreasing homophily and plateaus at values about 10%, for $\eta \leq 0.5$. Increasing the small world parameter from $\epsilon = 0.1$ (blue) to 0.5 (orange) reduces the small coverage effect even for high levels of homophily. Error bars show a standard deviation. **b** A parameter sweep over the homophily and small world parameters, $\eta$ and $\epsilon$, respectively, shows that the small coverage effect is most pronounced (close to 30%) for small $\epsilon$ and large $\eta$ and plateaus between 10 and 15% for other parameter settings.

sufficiently to trigger a resurgence of cases. This can be seen in Fig. 2d in the bifurcation of the infection risk, $IIR(M)$, at high coverage. In this regime, the epidemic either dies out quickly or larger outbreaks occur in both populations if a small number of infections persist until adherence has decreased sufficiently. One can think of this regime as an "all or nothing" scenario-driven by finite size effects in the network model. Consequently, the distribution of $IIR(M)$ values becomes bimodal in this regime with one mode being sharply distributed around zero (epidemic quickly dies out, lower limit of the CI) or at values much larger than zero (epidemic persists, upper limit). Figure 2d shows this as a bifurcation for high coverage values, $m > 0.6$.

In Fig. 2d, e we also show simulation results for a scenario where adherence does not wane, i.e. $q = 1$ (using $q_r(0) = q_s(0) = 0.4$). In this case, the frequently held belief of infection risk being a monotonously decreasing function of coverage, $m$, is recovered for both population groups and on an individual and population level. That is, the more people adopt protective behaviour, the smaller the outbreak size.

Note that the small coverage effect is not a result of the normalisation in the definition of the individual infection risks, $IIR(M = 1)$ and $IIR(M = 0)$. As the infection risk for adopting individuals increases with $m$ for small

coverage, Fig. 2d shows that the number of infections grows even faster than the number of adopting individuals in that group. The situation is slightly different on the population level, Fig. 2(e), where this effect is compensated by an increasing number of individuals moving from the non-adopting to the adopting state and thereby reducing their infection risk.

As seen in Fig. 2d, adopting behaviour at a coverage of 60% ($m = 0.6$) reduces the $IIR$ upon adopting protective behaviour from 95% for $M = 0$ to 78% for $M = 1$. For a coverage of 10% ($m = 0.1$), however, adoption reduces the risk to 61%. To explore this effect more systematically for different parameter settings, we consider the infection risk reduction (in percent) from a coverage range of 60% down to 10%. For the example given above, the small coverage effect thereby reduces the individual infection risk from 78% for $m = 0.6$ by 17% (absolute reduction) to 61% at $m = 0.1$.

The extent to which homophily amplifies the small coverage effect is shown in Fig. 3. Note that here we only show statistics from runs where the epidemic does not die out immediately at the beginning of the simulation, as these runs would inflate the CIs. For strong homophily ($\eta$ close to one), infection risk reduction falls within the range of 20% to 25%; see Fig. 3a. With $\eta$ decreasing, the infection risk reduction decreases too. For values of

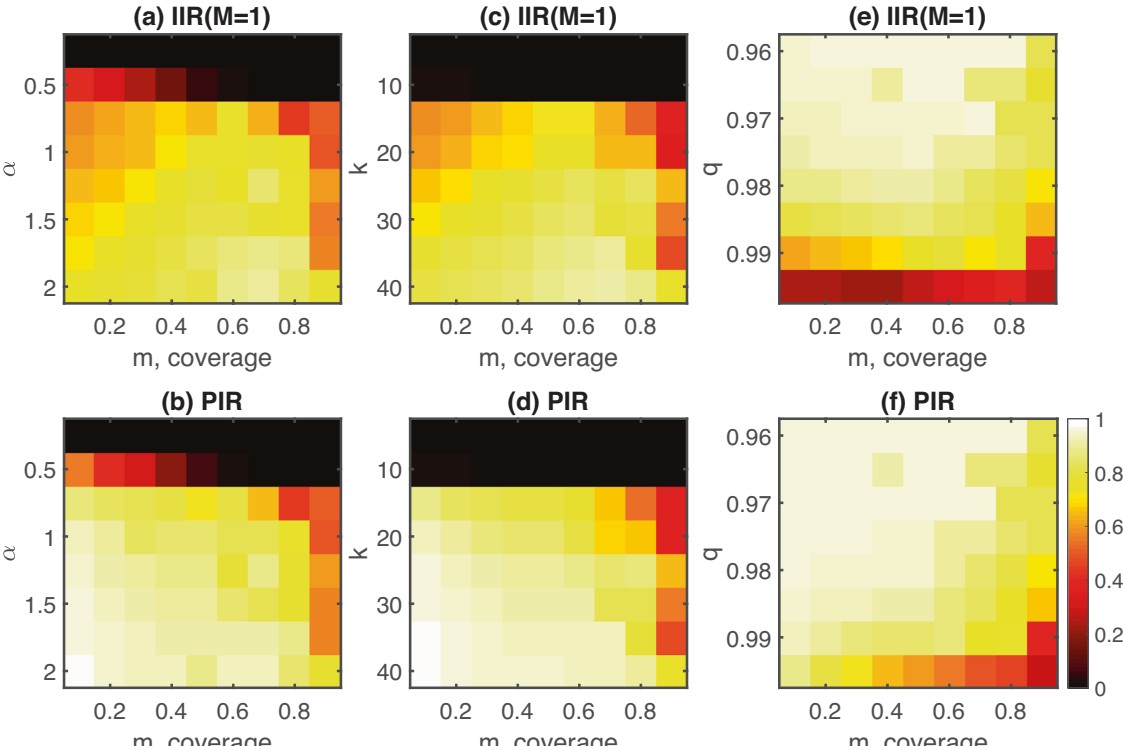

**Fig. 4 | Robustness of the small coverage effect.** For various values of the (**a**, **b**) transmission rate $\alpha$, (**c**, **d**) network degree $k$ and (**e**, **f**) waning parameter, $q$, we show the individual-level infection risk for adopters, $IIR(M = 1)$, and the population-level infection risk, $PIR$, respectively. For low values of $\alpha$ or $k$, outbreaks are suppressed and there is no small coverage effect. Above this suppression regime, we find a regime in which the effect can be observed with different magnitudes.

around $\eta \approx 0.5$ the infection risk reduction bottoms out at 10% where it assumes a plateau for weaker levels of homophily. Increasing the small world parameter $\epsilon$ to 0.5 decreases assortative mixing and leads to a smaller infection risk reduction for large $\eta$. These observations are corroborated by a parameter sweep over a range of values for $\eta$ and $\epsilon$; see Fig. 3b. Under strong assortative mixing (high $\eta$, small $\epsilon$), the small coverage effect reduces the infection risk by close to 30%. As assortative mixing decreases, the small coverage effect decreases and plateaus at ~10–15%.

Figure 4 shows results for the robustness of the small coverage effect with respect to different choices of the transmission rate $\alpha$, network degree $k$, and waning parameter $q$. Outbreaks are suppressed for low values of $\alpha$ or $k$, as shown by the small to vanishing individual-level and population-level infection risks in Fig. 4a–d. For values of $\alpha$ or $k$ above a certain threshold, large outbreaks can be observed (epidemic phase transition). Close to these thresholds, one can observe the largest differences in individual-level infection risk between very small and intermediate coverage. As $\alpha$ or $k$ increase further beyond these thresholds, the small coverage effect persists but becomes smaller in magnitude. A similar observation holds for decreasing waning parameters, see Fig. 4e, f.

So far, we have considered situations where the initial effectiveness of adopting protective behaviour is high ($q_r(0) = q_s(0) = 1$). Note that the small coverage effect can also be observed for less effective measures. For instance, for cloth masks the inward and outward protective effectiveness has been estimated as 50% and 30%, respectively[43]. Using these values ($q_r(0) = 0.5$, $q_s(0) = 0.3$), we observe infection risk reductions (from a coverage of 70% to 10%) of around 7% (SD 1.6%). One can also compare the relative risk ($RR$) of becoming infected between adopting and non-adopting individuals. For a coverage of 70%, adopting individuals had $RR = 0.95$ compared to non-adopting individuals, for small coverage ($m = 10$%) we found $RR = 0.88$. The magnitude of this reduction depends on the model parameters. For instance, setting the connectivity of the network to $k = 16$ gives a more pronounced infection risk reduction (at 10% compared to a coverage 90%) of 11% (SD 2.1%), meaning that the relative risks of becoming infected reduce from $RR = 0.95$ to 0.84.

## Discussion
Conventional wisdom holds that the more people wear a mask, the higher the protection it offers. In this study, we show that this is not necessarily true in general. Using the plausible assumptions that (i) adherence to protective behaviour wanes over time and (ii) individuals tend to have a preference to interact more with like-minded others (homophily), we demonstrate the existence of a hitherto, as far as we know, unknown small coverage effect: As the number of people adopting the protective behaviour increases, the individual-level protection offered by the behaviour decreases. This finding challenges the frequently held notion that the individual-level effectiveness of measures such as mask-wearing always increases with the number of people wearing a mask[44–49]. We numerically observed this effect for a wide range of parameters in a parsimonious network model. The small coverage effect can be observed independently from how strong individuals cluster topologically in the network (i.e., the friend of a friend is likely to be a friend of mine too) and in terms of adopting the behaviour (homophily). However, both types of clustering, when combined, do amplify the effect substantially.

The origin of the small coverage effect can be easily understood from system-dynamical properties. If only a small fraction of the population adopts protective behaviour, there is a good chance that the outbreak will already be over in the non-adopting population group before measure adherence has completely waned. The situation changes if there are more people initially adopting the behaviour. As the initial number of adopting individuals increases, so does the number of individuals that discontinue protective behaviour over a given time span. This means that the suscep-tibility in the population will grow faster as measure adherence is waning in a larger number of initially adopting individuals. This fuels–and thereby prolongs–the outbreak. Hence, the chance that the outbreak will be over before adherence has vanished completely decreases with coverage.

To illustrate this with a concrete example, consider a population of 1000 individuals and that 10 (1%) of them initially adopt protective behaviour. Assuming that the interventions are perfectly effective and that the waning rate is 10%, the number of susceptible individuals in the population increases by one in the first time step of the model. However, if the initial adoption rate is 90% or 900 individuals, the susceptible compartment would grow by 90 individuals in the first time step, almost doubling in size. This greater growth in the number of susceptible individuals increases the duration of the outbreak and, therefore, the number of initially adopting individuals who may still be exposed to the pathogen later in the simulation.

The small coverage effect, therefore, occurs when the duration of the outbreak size and the speed with which measure adherence wanes match each other in a certain relationship. Namely, the waning process must occur on a timescale where a sufficient number of individuals stop adhering while the number of infections is still high enough, i.e. the outbreak is still unfolding. The question then arises as to how exact this matching of the two timescales must be for the small coverage effect to occur. If the outbreak has already ended before there was substantial waning or if adherence has already completely waned before the outbreak even took off, one would expect a strongly diminished small coverage effect.

We find that the effect appears for a broad range of transmission rates, network connectivity, and waning rates of adherence. Specifically, there is no small coverage effect for parameter settings where the initial outbreak is suppressed (e.g., low transmission rates and/or low network connectivity) or where the waning rate is zero. Our simulations suggest that the maximum small coverage effect is observed above, but close to, the thresholds where initial outbreaks are no longer suppressed. If transmission rates or network connectivity increases even further, more infections have already occurred before substantial waning has taken place and the small coverage effect decreases in magnitude.

The small coverage effect also decreases in magnitude as the waning rate becomes larger ($q$ becomes smaller). The faster the waning process, the shorter the timespan in which the process can interact with the unfolding outbreak.

Homophily amplifies the small coverage effect. This is probably related to local trapping of infection events in the network[50,51]. Smaller groups of adopters that are preferentially connected to each other are less likely to be affected in the early stages of the outbreak, when adherence is still high. This is because each adopter is likely to be connected to others who are also adopting, thus protecting each other. They are also less likely to be affected by large numbers of infections in non-adopting individuals because they have fewer links with them. It will take longer for the pathogen to invade these groups of mutually protecting individuals than it would in a situation without homophily. Therefore, as homophily increases, there is a greater chance that the outbreak will be over before there has been any substantial decline in protective behaviour.

It has indeed been observed that adherence to protective measures such as wearing masks has strongly increased in the early phases of the pandemic[16,19,20] but has since declined[23,24]. Many studies that tried to measure the real-world effectiveness of mask-wearing basically compared the infection risk in adopting versus non-adopting individuals[52]. Systematic reviews have found that effectiveness can be quite heterogeneous across different settings and studies[10,53]. The reasons for this heterogeneity are not yet fully understood. The present work demonstrates that such study designs can be severely affected by the confounding influence of the population-wide mask coverage, which is typically not considered as a covariate. Using conservative estimates of the protective effectiveness of cloth masks, we find that the reduction in relative risk of masking versus non-masking individuals is between 5% and 15%. Further research is needed to understand whether empirically measured real-world differences in the effectiveness of mask-wearing could originate from the small coverage effect.

Academically, it would be challenging in the next step to demonstrate the mechanisms behind the small coverage effects also in the framework of game theory. One could think of a setting where adopters are cooperators and non-adopters are defectors; the payoff function (infection risks for adopters and non-adopters) would depend on the respective group sizes, and the waning effect would have to be implemented by a continuous shift towards a larger fraction of non-adopters over time. However, it is not clear from the outset what would constitute a round or interaction in which this game would be played. A conceivable option could be a model with waning immunity (i.e., transitions from the recovered to the susceptible state) and seasonal driving of the transmission risk, $\alpha$. Individuals could observe their risk of infection during an outbreak and possibly adjust their behaviour (state $M$) in the next seasonal outbreak depending on whether their past behaviour protected them or not.

Our simulation study is limited due to the use of a parsimonious model that makes idealised assumptions about several aspects, such as the infectious disease dynamics or the contact network topology. This means that certain real-world complexities, including variations in disease transmission dynamics and the intricacies of social contact networks, are not fully captured in our model. Specifically, our model does not account for factors such as variations in individual behaviours, population demographics, or the effects of different non-pharmaceutical interventions (NPIs) in mitigating disease spread. While this study design was a deliberate choice in order to isolate the mechanism driving the small coverage effect, it remains to be seen to what extent this effect is observed in more sophisticated and calibrated epidemiological models and, above all, in real-world settings.

Regarding the contact network topology, assortative mixing with regard to mask-wearing status has indeed been observed during "real world contacts"[33,34]. Furthermore, assortative mixing might arise due to mask mandates (e.g., in public transport, universities, etc.) in certain settings or due to other factors that confound mask-wearing status, as was observed for urban areas in Shanghai[54]. However, it is unclear to what extent this kind of assortative mixing actually occurs in all settings that are relevant sources of exposure for an individual, which in many cases may be the household setting. It is therefore unreasonable to assume a network topology with perfect homophily ($\eta = 1$) in real life situations. Note that the small coverage effect can also be observed to a lesser extent in situations with smaller or even no homophily ($\eta = 0$).

In terms of infectious disease dynamics, countries showed much more complex trajectories during the SARS-CoV-2 pandemic compared to the situations considered in this work. The small coverage effect can also be expected in scenarios or models with recurrent infection waves, provided that the probability to adopt protective behaviour increases again at the onset of a new wave (e.g., due to increased risk perception or mask recommendations/mandates being issued) before adherence wanes again. However, there is no reason to expect this effect in situations where incidence assumes a steady state with a roughly constant number of infections.

We emphasise that the small coverage effect only appears when looking at the individual-level infection risk. On a population level, we observe only minor changes in the infection risk for small and intermediate coverage values. This suggests that although individuals may experience reduced risk, the overall impact on the population's infection risk is not significant. There is a trend towards slightly decreasing population-level infection risks as a function of coverage, suggesting that increases in individual-level infection risks due to the small coverage effect are offset by the transmission-reducing effects of higher coverage. In other words, the collective impact of protective behaviours across the population helps to mitigate the spread of infection, even if some individuals experience a higher risk due to incomplete coverage. If coverage further increases and crosses a threshold where it becomes increasingly likely that outbreaks are completely suppressed, we observe a bifurcation in both the individual- and population-level infection risks. In this regime, outbreaks are either immediately suppressed (leading to a mode of outbreak sizes lying close to zero) or they are amplified due to waning adherence (giving a mode corresponding to high infection risks).

If it were true that wearing a mask offered less protection because fewer people were wearing masks, then it would be rational to abandon this behaviour because it wouldn't offer any protection if everyone else took off their masks. An important implication of our work is that quite the opposite

may be true for individual choices. Masking, when adopted by a minority, might actually facilitate sitting out a wave without becoming infected. However, at the population level, lower coverage leads to higher overall infection rates in our model. Therefore, our results should not be interpreted to mean that mask use is ineffective as a public health intervention. Rather, our results suggest that mask use may remain an effective intervention at the individual level even in situations where overall population coverage is low. Our results also highlight that the effectiveness of interventions might be assessed in a completely different way at the level of individuals (e.g., comparing the effectiveness of mask-wearing to prevent infections of individuals within the same population) or on the level of populations (comparing infection numbers between two populations with different coverage of mask-wearing). When interpreting studies assessing the effectiveness of such interventions, great care is needed to distinguish their individual- and population-level effectiveness and to consider the potential system-dynamical interplay between these two levels, such as the small coverage effect described in this work.

## Data availability
No new data were created or analysed in this study. The data underlying the resulting graphs and charts is accessible as a supplementary dataset.

## Code availability
Simulation code is at Zenodo[55].

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

## Author contributions

P.K. conceived the study. P.K. and K.L. performed the analyses. P.K. wrote the first draft of the article. K.L. and S.T. contributed to the writing. All authors reviewed and edited the manuscript.

## Competing interest

The authors declare no competing interests.
