## [Peer Review File · Communications Medicine]

Reviewers' comments:

Reviewer #1 (Remarks to the Author):

Thanks for the opportunity to review this interesting paper. The authors look at a structured network model of disease transmission to highlight the unintuitive finding that the more people engage in NPIs - up to a point - the higher their risk of infection will be. I thought the study was mostly well written, concise and the figures were clear.

I have some general comments and some minor specific points:

p3: "Consequently, the effectiveness of an NPI on the individual level (i.e., how much wearing a mask reduces one's personal risk of becoming infected) depends on system-dynamical properties such as the fraction of the entire population adopting the NPI. As a higher coverage rate moves the system closer to its epidemic threshold, higher population-level adoption typically also means a stronger individual-level infection risk reduction." - I don't think the logic of what the authors are stating here is obvious. As this is an important point in the paper, it would be helpful to provide more details as well as providing some previous studies to reference their claims.

Figure 1: caption should read "...and with another probability (η) the behaviour state (colour) is swapped..."

From p4 ("In the model...") this all seems like Methods to me, and would include them as such.

P7: "if none of the two" -> "if neither of the two"

Fig 2d: It's not totally clear to me why the confidence intervals are so wide in the simulations. My guess is that at high NPI coverage, outbreaks will not take off when the infection does not take off in the non-adopting group causing a bimodal outbreak size. Perhaps this could be made clearer.

Notation: Duration of infectiousness is denoted by β . As β is often used as the transmission rate, perhaps consider changing to γ as this is more widely used as duration of infectiousness.

General points:

1. The counterintuitive dynamic that is being discussed - as I understand it - relies on the two known phenomenon: i) if an intervention such as an NPI slows transmission, but not enough that $R_0 < 1$, then you expect a lower peak that occurs later than if the NPI had not been implemented, ii) As you increase the coverage of an intervention you increase the herd effects (up to a point at which effects start diminishing), therefore, iii) if you increase the coverage of an intervention you expect reduced individual level risk. So, there is a balance to be had between a good intervention that produces lots of herd effects but as a result delays the outbreak such that it has a lower effectiveness overall due to waning. Do the authors explain this reasoning somewhere in the paper? There is a bit about the mechanisms in the discussion but perhaps not sufficient.

2. I understand that the authors wish to keep a simple model to show the effect, however, there is very little uncertainty analysis that really convinces me that this can be an important phenomenon. For example, the study only looks at $R_0 = 2$, one effectiveness waning rate, and one NPI effectiveness. I think it's important that the authors show more comprehensively under what conditions this phenomenon holds. As I suggest above, whether the effect is observed will be a balance between the delay in the epidemic (caused by the size of the herd effects - a function of effectiveness of the NPI, strength of the homophily, number of contacts [this is not looked at in the model], and NPI coverage), and the speed at which the NPI effectiveness wanes. A more rigorous treatment of this would make a very nice paper.

3. I wanted to raise the issue of how relevant the concept of homophily is for NPIs such as mask wearing. I understand that my friends and family are the sort of people to likely e.g. wear a mask if I am the sort of person to wear a mask. However we typically won't be wearing a mask in each others company (in a household or when we mix with each other socially). In the cases where we are wearing masks it will be in a different setting e.g. on public transport etc. when the risks of transmission from others (not just my mask-wearing contacts) will be the main source of exposure. Therefore, I'd like to push back on the set up of the homophily and contact network set up within the paper. Thoughts from the authors on the use of their model would be welcome here.

4. I am not clear on whether individuals are set as adopters or non-adopters ($M = 0,1$) through the entire simulation (I think this is fixed, but could the authors clarify this). That is, is the IRR is calculated within the same population, regardless of whether the individuals from the adopting group are correctly adhering to the NPI? If by $t=200$, the intervention effectiveness is only 10%, it's probably because adopters aren't adhering to the intervention most of the time (or there's a 10% chance they totally adhere depending on how you assess it), therefore, is it really fair to calculate the IRR in an adopter population that is no longer adopting any protective behaviour?

Reviewer #2 (Remarks to the Author):

This paper shows that the infection protection conferred by mask wearing may, under some circumstances, reduce when the proportion of people wearing masks is greater. The authors refer to this as the "small coverage effect".

The small coverage effect occurs because when coverage is low the epidemic may burn out sufficiently quickly (due to the large number of unprotected individuals) to outweigh reducing adherence in agents who were initially mask-wearing.

Although this admittedly counterintuitive effect is of some interest, I don't think there is a huge contribution to either theory or real-world practice.

I had no problem with the reporting of the simulations; the paper is clearly written and appears technically competent (within the limits of my own expertise).

First, it could be argued that there is little practical relevance due to unrealistic assumptions. In the case of COVID19, for example, it is well established that the combination of vaccine waning, mutation, and increases then decreases in infection-induced immunity will result in a permanent level of infection, maybe between 2% and 5% of the population. Incorporation of this assumption might change the results substantially.

I appreciate the need (as noted in the discussion of limitations) to keep things simple. But perhaps the complexity is unavoidable for realism in the present case.

Second, in reality, the probability of adherence surely depends not just on the passage of time but also on prevailing levels of infection. If the epidemic lasts longer, decline in adherence will be slower, and vice versa. Again this (related points) is noted in the discussion, but in the real world it may be a major issue.

It would be good to have some analysis of population-level effects, for which the conclusions may be different.

As the authors note, previous studies have found mixed results in the effects of mask-wearing. The implication is that the mixed findings might reflect the small coverage effect. While I appreciate that

there are insufficient data to test this idea definitively, I suspect it should be possible to say something more about its plausibility based on real-world observations.

We thank the referees for their time and efforts to review our manuscript. We are glad to see that they saw merit in our approach. Their comments have been very helpful to further improve the manuscript. We have taken each of these comments thoroughly into account and changed the manuscript accordingly. In particular, we have carried out more extensive simulations and robustness tests and made major changes to how individual-level versus population-level effectiveness of the interventions is handled. Below, we provide a point-by-point response to each comment.

Reviewer #1 (Remarks to the Author):

Thanks for the opportunity to review this interesting paper. The authors look at a structured network model of disease transmission to highlight the unintuitive finding that the more people engage in NPIs - up to a point - the higher their risk of infection will be. I thought the study was mostly well written, concise and the figures were clear.

I have some general comments and some minor specific points:

p3: “Consequently, the effectiveness of an NPI on the individual level (i.e., how much wearing a mask reduces one’s personal risk of becoming infected) depends on system-dynamical properties such as the fraction of the entire population adopting the NPI. As a higher coverage rate moves the system closer to its epidemic threshold, higher population-level adoption typically also means a stronger individual-level infection risk reduction.” - I don’t think the logic of what the authors are stating here is obvious. As this is an important point in the paper, it would be helpful to provide more details as well as providing some previous studies to reference their claims.

Thank you for pointing this out. We have extended this part. We now introduce the concepts of “individual-level” and “population-level effectiveness” more explicitly in the introduction and give references from the literature that quantified one or the other. Natural experiments and ecological studies often measure population-level effectiveness while RCTs and case-control studies often measure individual-level effectiveness. We further provide references for works that studied the relationship between coverage and (mostly population-level effectiveness) within epidemiological models. We now state:

“Two types of effectiveness of NPIs can be distinguished for behavioural measures with imperfect (incomplete) adherence. First, by population-level effectiveness we refer to the overall reduction of infections when an NPI is adopted by a certain fraction of the population compared to a situation in which the NPI is adopted by a fewer number of people (e.g., zero) but everything else stays the same. Second, for the individual-level effectiveness we compare the infection risk for an individual that adopts an NPI to the risk of an individual not adopting the NPI within the *same* population. Studies that quantify effectiveness by comparing the number of infections in regions with and without an NPI seek to estimate population-level effectiveness, as it is often the case in ecological study designs or studies relying on natural experiments (Flaxman 2020, Haug 2020, Sharma 2021, Liu 2021). Case-control studies (Lio 2021, Doung-Nern 2020) or randomized controlled trials (Bundgaard 2021) often measure individual-level effectiveness.

Epidemiological models make it clear that individual-level and population-level effectiveness are very different concepts. It is well-established that if the rate of adoption (coverage rate) and the effectiveness of NPIs are high enough, epidemic outbreaks can be successfully suppressed, meaning that a primary infection will lead to (on average) less than one secondary cases (Eikenberry 2020, Ma 2021, Rao 2021). In this case of suppression an NPI can have a high population-level effectiveness combined with low individual-level effectiveness, as non-adopters are indirectly protected through the overall low infection risk. For lower coverage, where no suppression of the outbreak occurs, population-level effectiveness might be low but individual-level effectiveness might be much higher than in situations where outbreaks are suppressed. In other words, the effectiveness of an NPI on the individual level (i.e., how much wearing a mask reduces one's personal risk of becoming infected) depends on population-level properties such as the fraction of the entire population adopting the NPI."

Figure 1: caption should read "...and with another probability ($1-\eta$) the behaviour state (colour) is swapped..."

Thank you for catching this, corrected.

From p4 ("In the model...") this all seems like Methods to me, and would include them as such.

Yes, moved to methods.

P7: "if none of the two" -> "if neither of the two"

Corrected.

Fig 2d: It's not totally clear to me why the confidence intervals are so wide in the simulations. My guess is that at high NPI coverage, outbreaks will not take off when the infection does not take off in the non-adopting group causing a bimodal outbreak size. Perhaps this could be made clearer.

For coverage values above a certain threshold, there indeed emerges a bimodal distribution of outbreak sizes, with very sharp peaks. To show this more clearly, we have shown this "bifurcation" now in the plots. In the results section we have added:

"Consequently, the distribution of IIR(M) values becomes bimodal in this regime with one mode being sharply distributed around zero (epidemic quickly dies out, lower limit of the CI) or at values much larger than zero (epidemic persists, upper limit). Figure 2(d) shows this as a bifurcation for high coverage values, $m > 0.6$."

And in the caption of Figure 2:

If measure adherence wanes, the infection risk increases with m for $m < 0.6$; for higher values of m the distribution of outbreak sizes becomes bimodal (marker size is proportional to the fraction of simulation runs in each mode). The infection risk is always a monotonously decreasing function of m if adherence does not wane. On the (e) population level we observe a plateau with similar

population infection risks, β , for a wide range of intermediate coverage and also a bifurcation for high coverage.”

Notation: Duration of infectiousness is denoted by $1/\beta$. As β is often used as the transmission rate, perhaps consider changing to $1/\gamma$ as this is more widely used as duration of infectiousness.

We replaced β with γ .

General points:

1. The counterintuitive dynamic that is being discussed - as I understand it - relies on the two known phenomenon: i) if an intervention such as an NPI slows transmission, but not enough that $R_0 < 1$, then you expect a lower peak that occurs later than if the NPI had not been implemented, ii) As you increase the coverage of an intervention you increase the herd effects (up to a point at which effects start diminishing), therefore, iii) if you increase the coverage of an intervention you expect reduced individual level risk. So, there is a balance to be had between a good intervention that produces lots of herd effects but as a result delays the outbreak such that it has a lower effectiveness overall due to waning. Do the authors explain this reasoning somewhere in the paper? There is a bit about the mechanisms in the discussion but perhaps not sufficient.

Thank you for this suggestion. Also in line with a comment from another reviewer, we extended this discussion and added the following paragraphs in the discussion:

“The small coverage effect therefore occurs when the duration of the outbreak size and the speed with which measure adherence wanes match each other in a certain relationship. Namely, the waning process needs to happen on a timescale on which a sufficient number of individuals stops adhering while infection numbers are high enough, i.e., the outbreak is still unfolding. The question then arises how “exact” this matching of these two timescales needs to be for the small coverage effect to emerge. If the outbreak has already ended before there was substantial waning or if adherence has already completely waned before the outbreak even took off, one would expect a strongly diminished small coverage effect.

We find that the effect appears for a broad range of transmission rates, network connectivities, and waning rates for the adherence. More concrete, there is no small coverage effect for parameter settings where the initial outbreak is suppressed, (e.g., low transmission rates and/or low network connectivity) or where the waning rate is zero. Our simulations suggest that above but close to the thresholds where initial outbreaks are not suppressed anymore, the maximal small coverage effect can be observed. If transmission rates or network connectivity increases even further, more infections have already occurred before substantial waning has taken place and the small coverage effect decreases in magnitude.

The small coverage effect also decreases in magnitude as the waning rate becomes larger (η becomes smaller). The faster the waning process, the smaller the timespan in which the process can “interact” with the unfolding outbreak.”

2. I understand that the authors wish to keep a simple model to show the effect, however, there is very little uncertainty analysis that really convinces me that this can be an important

phenomenon. For example, the study only looks at $R_0 = 2$, one effectiveness waning rate, and one NPI effectiveness. I think it's important that the authors show more comprehensively under what conditions this phenomenon holds. As I suggest above, whether the effect is observed will be a balance between the delay in the epidemic (caused by the size of the herd effects - a function of effectiveness of the NPI, strength of the homophily, number of contacts [this is not looked at in the model], and NPI coverage), and the speed at which the NPI effectiveness wanes. A more rigorous treatment of this would make a very nice paper.

As described above, we have substantially expanded the discussion of how comprehensively the phenomenon holds. We have also added robustness tests in a new figure and describe them in results:

"Figure 4 shows results for the robustness of the small coverage effect with regard to different choices of the transmission rate β , network degree k , and waning parameter γ . Outbreaks are suppressed for low values of β or k , as shown by the small to vanishing individual-level and population-level infection risks in Figure 4(a-d). For values of β or k above a certain threshold, large outbreaks can be observed (epidemic phase transition). Close to these thresholds, one can observe the largest differences in individual-level infection risks between very small and intermediate coverage. As β or k increase further beyond these thresholds, the small coverage effect persists but becomes smaller in magnitude. A similar observation holds for decreasing waning parameters, see Figure 4."

3. I wanted to raise the issue of how relevant the concept of homophily is for NPIs such as mask wearing. I understand that my friends and family are the sort of people to likely e.g. wear a mask if I am the sort of person to wear a mask. However we typically won't be wearing a mask in each others company (in a household or when we mix with each other socially). In the cases where we are wearing masks it will be in a different setting e.g. on public transport etc. when the risks of transmission from others (not just my mask-wearing contacts) will be the main source of exposure. Therefore, I'd like to push back on the set up of the homophily and contact network set up within the paper. Thoughts from the authors on the use of their model would be welcome here.

Thank you for emphasizing this interesting point. There are indeed a number of studies that show assortative mixing also in real life community contacts. We describe this literature now in more detail in the introduction:

"A study evaluating mask-wearing status of pairs of people using public livestream footage from webcams across seven US cities in September 2020 found that 83% of the participants had the same status than the person they had contact with (Woodcock 2021). In 41% (42%) of the cases both individuals both were (not) wearing masks, respectively, confirming strong assortative mixing patterns in social contexts. Observational studies in the Czech Republic confirmed these observations by showing that mask-wearing status of other customers in a shop was one of the strongest predictors for mask-wearing behaviour of new customers (Mladenovic 202)."

As we agree with the assessment of the referee that for real life contact networks perfect assortative mixing is a highly unrealistic assumption, we have added a discussion of this point under limitations:

"Regarding the contact network topology, assortative mixing with regard to mask-wearing status has indeed been observed during "real world contacts" (Woodcock 2021, Mladenovic 2023). Furthermore, assortative mixing might arise due to mask mandates (e.g., in public transport, universities, etc.) in certain settings or due to other factors that confound mask-wearing status, as

was observed for urban areas in Shanghai (English 2021). Still, it is unclear to which extent this kind of assortative mixing indeed occurs in all settings that are relevant sources of exposure for an individual, which in many cases might be the household setting. It is therefore unreasonable to assume a network topology with perfect homophily ($\eta = 1$) in real life situations. Note that the small coverage effect can also be observed with reduced magnitude in situations with smaller or even without homophily ($\eta = 0$).

4. I am not clear on whether individuals are set as adopters or non-adopters ($M = 0,1$) through the entire simulation (I think this is fixed, but could the authors clarify this). That is, is the IRR is calculated within the same population, regardless of whether the individuals from the adopting group are correctly adhering to the NPI? If by $t=200$, the intervention effectiveness is only 10%, it's probably because adopters aren't adhering to the intervention most of the time (or there's a 10% chance they totally adhere depending on how you assess it), therefore, is it really fair to calculate the IRR in an adopter population that is no longer adopting any protective behaviour?

To clarify that the set of adopters remains static, we changed the beginning of Methods to:

"In the model, all individuals have a static state that remains fixed throughout the simulations indicating whether they are willing to adopt protective behaviour or not. Over time, however, the probability by which they indeed adopt this behaviour decreases over time to implement waning of NPI adherence."

We implement the waning rather slow (compared to the outbreak sizes, see above) as otherwise, if the waning would take place on a completely different time scale, no such effect could be observed. Hence, IRR is indeed computed within an adopting population in which the effectiveness of the intervention decreases due to decreased adherence (but in the main results shown it is not true that they do not show any protective behaviour at all). We hope that the extended discussion of the timescale effects given above further clarifies this.

It also worth noting that the coverage effect increases with clustering (smaller parameter ϵ). If the set of adopters would be randomized at each run, this would "destroy" this clustering of adopters and lead to a different model. As the adopting population would not be fixed in this hypothetical case, it wouldn't make sense to define an IRR for adopters. Hence, we do not consider such situations in the current work.

Reviewer #2 (Remarks to the Author):

This paper shows that the infection protection conferred by mask wearing may, under some circumstances, reduce when the proportion of people wearing masks is greater. The authors refer to this as the "small coverage effect".

The small coverage effect occurs because when coverage is low the epidemic may burn out sufficiently quickly (due to the large number of unprotected individuals) to outweigh reducing adherence in agents who were initially mask-wearing.

Although this admittedly counterintuitive effect is of some interest, I don't think there is a huge contribution to either theory or real-world practice.

I had no problem with the reporting of the simulations; the paper is clearly written and appears technically competent (within the limits of my own expertise).

We thank the referee for the time and effort invested to review our manuscript and thereby helping us to further improve it. We have made several major changes in response to these comments. In particular, we seek to clarify and adjust the main (practical) message of our manuscript. What we show is that – for a broad range of parameters – assessing the effectiveness of certain types of intervention might lead to different conclusions when assessed on an individual or population level. In our view, mathematical models have so far mostly focussed on population-level effects and typically do not consider individual-level effects, even though the latter are often measured in RCTs or case-control studies. Second, we acknowledge that there is insufficient data to make any informed statement regarding how relevant this effect was in a particular situation. Instead, we now report additional results and discussion regarding the regime under which this effect can be observed, along with a discussion of how this relates to the SARS-CoV-2 pandemic. More details are given in the responses below.

First, it could be argued that there is little practical relevance due to unrealistic assumptions. In the case of COVID19, for example, it is well established that the combination of vaccine waning, mutation, and increases then decreases in infection-induced immunity will result in a permanent level of infection, maybe between 2% and 5% of the population. Incorporation of this assumption might change the results substantially.

I appreciate the need (as noted in the discussion of limitations) to keep things simple. But perhaps the complexity is unavoidable for realism in the present case.

We have added a discussion of this point now in more detail. When describing the limitations we have added:

“Regarding the infectious disease dynamics, countries have shown much more complex trajectories during the SARS-CoV-2 pandemic compared to the situations considered in this work. The small coverage effect can also be expected in scenarios or models with recurrent infection waves, provided that the probability to adopt protective behaviour increases again at the onset of a new wave (e.g., due to increased risk perception or mask recommendations/mandates being issued) before adherence wanes again. In situations where incidences assume a steady state with roughly constant numbers of infections, however, there is no reason to expect this effect.”

As a comment, in the early stages of this work we indeed used a SIRS model with seasonal driving and observed under the assumptions outlined above indeed also a small coverage effect. The isolation of this effect becomes, however, much more involved due to the additional parameters and mechanisms (e.g., immunity waning). And even then the model would still have to be considered as highly unrealistic. Hence, we chose to go for an even more parsimonious setup in which we can properly isolate the effect. Furthermore, for COVID-19 it is questionable whether we can expect a steady state of infections in regions with strong seasonal forcing. Wastewater-based surveillance for Austria, at least, does not suggest the emergence of a steady state (yet). We have, however, stated now explicitly that the mechanisms shouldn't be at work in situations of steady infection levels.

Second, in reality, the probability of adherence surely depends not just on the passage of time but also on prevailing levels of infection. If the epidemic lasts longer, decline in

adherence will be slower, and vice versa. Again this (related points) is noted in the discussion, but in the real world it may be a major issue.

We have added robustness tests to address this “matching of timescales” issues and show that there is indeed a wider range of parameter settings, under which the effect emerges. We describe these tests in the results:

“Figure 4 shows results for the robustness of the small coverage effect with regard to different choices of the transmission rate α , network degree k , and waning parameter q . Outbreaks are suppressed for low values of α or k , as shown by the small to vanishing individual-level and population-level infection risks in Figure 4(a-d). For values of α or k above a certain threshold, large outbreaks can be observed (epidemic phase transition). Close to these thresholds, one can observe the largest differences in individual-level infection risks between very small and intermediate coverage. As α or k increase further beyond these thresholds, the small coverage effect persists but becomes smaller in magnitude. A similar observation holds for decreasing waning parameters, see Figure 4.”

And discuss the implications of these tests for when the effect can be expected to occur in the discussion:

“The small coverage effect therefore occurs when the duration of the outbreak size and the speed with which measure adherence wanes match each other in a certain relationship. Namely, the waning process needs to happen on a timescale on which a sufficient number of individuals stops adhering while infection numbers are high enough, i.e., the outbreak is still unfolding. The question then arises how “exact” this matching of these two timescales needs to be for the small coverage effect to emerge. If the outbreak has already ended before there was substantial waning or if adherence has already completely waned before the outbreak even took off, one would expect a strongly diminished small coverage effect.

We find that the effect appears for a broad range of transmission rates, network connectivities, and waning rates for the adherence. More concrete, there is no small coverage effect for parameter settings where the initial outbreak is suppressed, (e.g., low transmission rates and/or low network connectivity) or where the waning rate is zero. Our simulations suggest that above but close to the thresholds where initial outbreaks are not suppressed anymore, the maximal small coverage effect can be observed. If transmission rates or network connectivity increases even further, more infections have already occurred before substantial waning has taken place and the small coverage effect decreases in magnitude.

The small coverage effect also decreases in magnitude as the waning rate becomes larger (η becomes smaller). The faster the waning process, the smaller the timespan in which the process can “interact” with the unfolding outbreak.”

It would be good to have some analysis of population-level effects, for which the conclusions may be different.

Results for the population-level were reported by means of the “population-level infection risk” (PIR). We have expanded the introduction to clarify the dichotomy between PIR and IIR. Further results for the PIR are now also shown in Figure 4 (see above), where we show both types of risk for wide ranges of parameters. We further emphasize now in the discussion the difference of individual- and population-level effects:

“We emphasize that the small coverage effect only appears when looking at the individual-level infection risk. On a population level we observe only minor changes in the infection risk for small and intermediate coverage values. There is a trend towards slightly decreasing population-level infection risks as a function of coverage, suggesting that increases in individual-level infection risks due to the small coverage effect are offset by the transmission-reducing effects of higher coverage. If coverage further increases and crosses a threshold where it becomes increasingly likely that outbreaks are completely suppressed, we observe a bifurcation in both the individual- and population-level infection risks. In this regime, outbreaks are either immediately suppressed (leading to a mode of outbreak sizes lying close to zero) or they get amplified due to waning adherence (giving a mode corresponding to high infection risks).”

As the authors note, previous studies have found mixed results in the effects of mask-wearing. The implication is that the mixed findings might reflect the small coverage effect. While I appreciate that there are insufficient data to test this idea definitively, I suspect it should be possible to say something more about its plausibility based on real-world observations.

As outlined in the previous responses, we have made several major changes to allow readers to better understand under which circumstances the effect can be expected to occur. We reflect this also now when making the conclusions of our work, which we have changed to the following as the main takeaway (along with similar changes in the abstract):

“Our results demonstrate that the effectiveness of interventions might be assessed in a completely different way on the level of individuals (e.g., comparing the effectiveness of mask-wearing to prevent infections of individuals within the same population) or on the level of populations (comparing infection numbers between two populations with different coverage of mask-wearing). When interpreting studies assessing the effectiveness of such interventions, great care is needed to distinguish their individual- and population-level effectiveness and to consider the potential system-dynamical interplay between these two levels, such as the small coverage effect described in this work.”

REVIEWERS' COMMENTS:

Reviewer #1 (Remarks to the Author):

Thank you for responding so well to my comments. I have no further major comments. However, there are a few minor typos throughout:

Abstract: repeated phrase on last line

Intro: 41% (42%) looks like a typo

p4: 'a fewer number of people' -> 'fewer people'

p4: 'less than one secondary cases' - 'fewer than 1 secondary case'

Fig 2 caption: spill over of text

p20: 'more concrete' -> 'more concretely'

Reviewer #2 (Remarks to the Author):

This paper is clearly improved; the distinction between PIR and IIR is much more explicit and this does help.

The authors have also added in a number of appropriate robustness checks and qualifications.

I did still feel that it is too difficult - especially in the abstract/introduction summaries - to understand at an intuitive level just why the effect is occurring. It's also not clear (in the summaries) what the role of homophily is, i.e., the extent to which the effect relies on it and why.

So: I still think the abstract and intuitive explanations need more work of this paper is to be accessible to a broad audience within a reasonable amount of reading time. Even after spending a lot longer on it than the average reader would be able to, and devote being reasonably scientifically literature, I am still not confident that I could explain this paper to another person.

We thank the for reviewing our manuscript again and for providing constructive remarks. We are also glad to see that they found the revised manuscript to be improved. We have again revised the manuscript to address the remaining issues, see below.

Reviewer #1 (Remarks to the Author):

Thank you for responding so well to my comments. I have no further major comments. However, there are a few minor typos throughout:

Abstract: repeated phrase on last line

Intro: 41% (42%) looks like a typo

p4: 'a fewer number of people' -> 'fewer people'

p4: 'less than one secondary cases' - 'fewer than 1 secondary case'

Fig 2 caption: spill over of text

p20: 'more concrete' -> 'more concretely'

Thank you very much, we implemented these changes.

Reviewer #2 (Remarks to the Author):

This paper is clearly improved; the distinction between PIR and IIR is much more explicit and this does help.

The authors have also added in a number of appropriate robustness checks and qualifications.

I did still feel that it is too difficult - especially in the abstract/introduction summaries - to understand at an intuitive level just why the effect is occurring. It's also not clear (in the summaries) what the role of homophily is, i.e., the extent to which the effect relies on it and why.

So: I still think the abstract and intuitive explanations need more work of this paper is to be accessible to a broad audience within a reasonable amount of reading time. Even after spending a lot longer on it than the average reader would be able to, and devote being reasonably scientifically literature, I am still not confident that I could explain this paper to another person.

Thank you for pointing out the need to improve the accessibility of the manuscript. We have revised the abstract to make it more readable (though we are afraid there is only so much one can do in explaining system-level dynamical phenomena in 250 words) and added a plain language summary per editorial request. We have also extended the discussion to make the mechanisms at work more transparent, also regarding homophily. For instance, we have added a small numerical example that should make it clearer for readers why the effect occurs:

“To illustrate this with a concrete example, consider a population of 1,000 individuals and that ten (1%) of them initially adopt protective behaviour. Assuming that the interventions are perfectly effective and that the adoption rate is 10%, the number of susceptible individuals in the population increases by one in the first time step of the model. However, if the initial adoption rate is 90% or

900 individuals, the susceptible compartment would grow by 90 individuals in the first time step, almost doubling in size. This greater growth in the number of susceptible individuals increases the duration of the outbreak and therefore the number of initially adopting individuals who may still be exposed to the pathogen later in the simulation.”

and

“Homophily amplifies the small coverage effect. [...] Smaller groups of adopters that are preferentially connected to each other are less likely to be affected in the early stages of the outbreak, when adherence is still high. This is because each adopter is likely to be connected to others who are also adopting, thus protecting each other. They are also less likely to be affected by large numbers of infections in non-adopting individuals because they have fewer links with them. It will take longer for the pathogen to “invade” these groups of mutually protecting individuals than it would in a situation without homophily. Therefore, as homophily increases, there is a greater chance that the outbreak will be over before there has been any substantial decline in protective behaviour.”